# Applications of PERALS-Alpha Spectrometry for the Investigation of Radionuclides in Water Samples

**DOI:** 10.3390/ma14143787

**Published:** 2021-07-06

**Authors:** Markus Zehringer, Franziska Kammerer, Anja Pregler

**Affiliations:** 1Alpweg 8, 4132 Muttenz, Switzerland; 2State-Laboratory of Basel-City, Kannenfeldstrasse 2, 4056 Basel, Switzerland; franziska.kammerer@bs.ch (F.K.); anja.pregler@bs.ch (A.P.)

**Keywords:** PERALS, alpha spectrometry, polonium, radon, radium, radiostrontium, thorium, uranium, actinides

## Abstract

In this paper, experiences of the last 20 years with the PERALS-technique are described. PERALS stands for photo electron-rejecting alpha liquid scintillation. This liquid scintillation technique was developed by Jack McDowell in the 1970s and is a powerful technique for the analyses of many natural alpha nuclides and also the beta nuclide ^90^Sr. The principle is based on a selective extraction of the radionuclide from the water phase by means of a complexing or ion pair reagent. The extractant contains also a cocktail suitable for scintillation counting. Therefore, the extract can be analyzed directly after the extraction step. After removing quenchers, such as oxygen, and the proper setting of a pulse shape discriminator, alpha pulses can be counted with a photomultiplier. This paper describes the development of robust analysis schemes for the determination of traces of polonium, thorium, uranium and other actinides in water samples (groundwater, rain water, river water, drinking water, mineral water, sea water). For radon and radium, the enrichment in the extract is poor. Therefore, PERALS methods are not suitable for trace analyses of these analytes. In addition, the extraction of the beta-emitter ^90^Sr with a PERALS cocktail is discussed, even though its beta spectrum is not analyzed with a PERALS counter. Results from the survey of drinking water and mineral water in Switzerland are presented for every radio element.

## 1. Introduction

### 1.1. The Legal Basis in Switzerland

Drinking water is the most important source of nourishment and therefore needs special care and analysis. Besides many possibilities of contamination, natural radionuclides can also occur in drinking water and can cause higher ingested doses by its regular consumption. The mean consumption is two liters per day or 730 L per year. Special care is needed for children. Some months before and after birth, the fetus or the baby, respectively, is more sensitive to radiation, especially when applied in low doses. The International Commission for Radiological Protection (ICRP) tries to respect this by proposing higher dose conversion factors for infants [1]. In Germany, limits are lower when drinking water is used to feed babies and to prepare infant formulas (see Table 1) [2]. Using the consumption rate of 730 L, an annual dose limit of 0.1 mSv and the corresponding dose coefficient e_ing_, a derived activity level (DAL) for each radionuclide can be calculated as follows:DAL (Bq/L) = 100 (μSv)/(730 (L) × eing (µSv/Bq))(1)

Since 2017, natural radionuclides have been regulated in the *Swiss Ordinance on drinking water and water for public baths and shower facilities* (TBDV) [3]. The limits in this ordinance were adopted from *Council Directive 2013/51/EURATOM* [4].

Unfortunately, only drinking water is regulated in the TBDV. For mineral waters or other beverages no more regulations in Switzerland have been added since the abrogation of the *Ordinance on Contaminants and Constituents in Food* [6]. Fortunately, there are efforts being made to adopt the Swiss Ordinance TBDV for mineral waters and other drinks as well. Nevertheless, a regular monitoring of drinking water, mineral waters and all other drinks (wine, beer, etc.) is essential. Therefore, robust and adequate equipment and practical experiences are necessary for analytical laboratories, which are involved in the radiological survey of drinking water.

Ground and spring water may be contaminated with natural radionuclides when passing certain rock and soil formations. Figure 1 shows the concentrations of uranium in drinking water in Switzerland. Heinz Surbeck, a Swiss pioneer in alpha spectrometry, has collected activity data of natural radionuclides in groundwater and tap water of Switzerland (around 3000 water samples) over the last few years. As expected, the plot refers to the geology of the country. The highest uranium concentrations (red dots) are found in the Alps (granitic rock). From 1990 on, data exist from different regions of Switzerland, but no systematic survey of the whole of Switzerland has been performed yet. Surbeck’s survey was published in the yearly monitoring report of the Federal Office of Public Health [7].

In this chapter, the experiences of water laboratories and our own experiences in the applications of alpha spectrometry for the analyses of water samples are represented. They are focused on the alpha-spectrometric technique, which was developed by Jack McDowell and Gerald Case, and is referred to as PERALS spectrometry.

### 1.2. The PERALS Method

The name PERALS stands for photo emission-rejecting alpha liquid spectrometry and it was developed in the 1970s by W. Jack McDowell and Gerald N. Case at the Oak Ridge National Laboratory, USA. McDowell was involved in separation science and therefore could incorporate his experiences in solvent extraction techniques. This involves the combination of specific solvent extraction and pulse shape analysis (PSA), which resulted in the development by other companies of electronic circuits to separate alpha pulses: pulse shape analysis (PSA) by Wallac Oy or pulse decay analyses (PDA) by Packard Instruments. They all take advantage of the longer pulse length that is produced by alpha particles compared to the pulse length of the beta particle. The comparison of the total pulse area with the pulse area after 50 ns makes it possible to categorize a pulse as alpha or beta. The PERALS spectrometer allows one to set this discrimination between alpha and beta/gamma for each sample extract by means of a pulse shape discriminator (PSD). The discrimination of beta and gamma pulses results in a remarkable reduction of the spectral background and an improvement of the energy resolution (~100 keV). This resolution is not sufficient for natural samples, containing different alpha nuclides. To overcome this problem, the alpha nuclides are extracted specifically from the sample using different extractants. The extractant is dissolved in a water immiscible solvent (e.g., toluene) together with a fluor (scintillator) and an energy transfer agent.

In concluding, the success of PERALS involves the enrichment of radionuclides by a specific extraction procedure using a water-immiscible solvent in combination with a fluor and an energy transfer agent and the PSD technique to count the alpha decays emitted from these extracts. The basic reference of PERALS is Jack and Betty McDowell [8] and also the numerous technical papers they published together with G.N. Case [9,10,11]. M. Kopp and D.M. Kopp from ORDELA Inc. finally improved the PERALS methods and presented a data map software to display alpha and beta/gamma counts as a pulse shape versus pulse height plot [12]. Scientists successfully introduced the PERALS technique in their laboratories (e.g., [13,14,15]). The PERALS spectrometer is well described in the publication of McDowell [8] and in the instruction manual for the PERALS Spectrometer from ORDELA Inc. [16] (Figure 2).

### 1.3. Selective Extraction

McDowell and Case developed specific extractant/scintillator solutions for actinides, which are commercially available from ORDELA Inc. (Table 2). The name of the extraction solution (cocktail) stands for the elements that can be analyzed with it. ALPHAEX is a universal extractant for many actinides and lanthanides. RADAEX and STRONEX, extraction cocktails for radium and strontium, contain lipophilic acids and are stored in acidic form. For their proper use, they have to be converted into their ionic form by mixing with sodium hydroxide and sodium nitrate.

### 1.4. General Procedure for the Extraction of Radionuclides from Water Samples

Over the last few years, scientists of the state laboratory of Basel–City, Switzerland, have developed very sensitive analytical methods for the determination of alpha nuclides in water samples. The proposed methods of Jack McDowell were adapted for the extraction of high water volumes (e.g., 1 L samples). Methods for polonium, strontium, thorium and uranium were implemented and successfully used in the daily control of drinking water. The extraction from high-volume samples proved to be unsuitable for radium and radon due to the low distribution coefficients (D) of the extractants RADAEX and RADONS. To achieve a sufficient enrichment by extraction of 1 L samples with some mL of extractant, partition coefficients should be >1000 [8].

The recovery of extraction can be calculated as follows (2), according to [8,17]:R (%) = ((D × V_org_)/(D × V_org_ + V_Aq_)) × 100(2)
with V_org_—volume of the extraction solution, V_Aq_—volume of the water sample, D—partition coefficient.

Example: with a partition coefficient D of 6000 of STRONEX, the theoretically achievable extraction recovery of a unique extraction of 1 liter of water with 8 mL of STRONEX should be 98% for ^90^Sr. Of course, the extraction is also influenced by other factors, such as the ionic strength of the water sample, the concentration of counter ions, pH value, etc.

For the procedure of extractions, only a few materials have to be available: 1 L glass bottles and phase separators fitting on the glass bottles (Figure 3), a magnetic stirrer with magnetic bars, reagent vials and 2 mL culture tubes. Before starting the extraction, the glass bottles have to be cleaned with acid solution to avoid any memory effects or cross contaminations [16]. In the state laboratory of Basel–City, the cleaned bottles are rinsed three times with 10 mL of 2 M acid solution (hydrochloric acid or nitric acid). Finally, the bottles are washed with distilled water. Another possibility is the automatic cleaning of the glassware using hot acid vapors.

All used chemicals should be of the purest quality possible, e.g., inorganic concentrated acids of suprapure quality.

One liter of water sample is conditioned for extraction as recommended by ORDELA [8,18], and then extracted with the chosen PERALS extractive scintillator solution for five minutes (stirring with magnetic bar). After complete phase separation, the organic layer is raised and drawn into the phase separator by adding distilled water via the funnel A (Figure 3, right). The extract is transferred into a reagent vial and purged with argon for at least 30 s. Finally, 1.5 mL of this solution is pipetted into a 2 mL culture tube (available from Schott (Mainz, Germany), ORDELA (Oak Ridge, TN, USA, etc.) which is then counted with a PERALS spectrometer. In the case of strontium analyses, the whole extract is counted with a conventional alpha/beta liquid scintillation counter.

### 1.5. Alpha Spectrometry, PSD

Coextracted ions, molecules with electronegative substituents (such as CCl_4_, etc.), oxygen and any color in the extract may cause a quenching of the alpha spectrum. Peak heights are reduced and the peaks are shifted to lower energies. Oxygen is removed by purging the extract with argon gas. Other quenchers cannot be eliminated. Possibly, colored extracts can be discolored with hydrogen peroxide. That means that quenching occurs in each alpha spectrum of the real samples. Therefore, stocked samples (with added analytes of a well-known activity) have to be analyzed in the series. Via the counting of such a spiked sample extract in the pulse shape mode, the PSD can be set correctly. Figure 3 shows the pulse shape spectrum of a ^226^Ra standard (available from ORDELA Inc., Oak Ridge, TN, USA). The left peak gives the sum of all beta and gamma pulses. These have to be eliminated from the counting process. The PSD is set between the beta/gamma and the alpha peak. This separation of the beta/gamma and alpha counts is less pronounced in real sample extracts. Sometimes three peaks, i.e., separate alpha, beta and gamma peaks, are observed. Therefore, it is advisable to analyses first a spiked sample extract with a higher activity of radionuclide. In general, the PSD remains constant for samples of the same matrix, but may slightly change between different PERALS counters. Therefore, the proper PSD has to be set for every used PERALS counter.

Trace analysis needs a longer counting time. Normally, the counting time of samples with 1 liter volume is 24 h. Sometimes, the quenching of the samples may vary. For the identification of the radionuclides, it may be helpful to overlay the spectrum with the spectrum of a spiked sample. For example, the software Interwinner from ITECH, which is used at the state laboratory of Basel–City, allows us to overlay spectra and to eliminate different forms of quenching by shifting the spectra on the energy scale [21]. This helps to identify radionuclides, which is sometimes difficult due to the low activity of concentrations.

For each radioelement, the conditions and extractive scintillators, as well as counting parameters, are different. These are described on the following pages together with the results of the monitoring of Swiss tap water and mineral waters.

## 2. Polonium

All nuclides of polonium, from ^190^Po up to ^220^Po, are unstable. The radiologically relevant polonium species belong to the natural decay series. These are ^218^Po, ^214^Po and ^210^Po, members of the ^238^U-decay chain, ^216^Po and ^212^Po from the ^232^Th decay chain and ^215^Po and ^211^Po from the ^235^U-decay chain. Most of them have very short half-lives, from microseconds to minutes. Only ^210^Po, with a half-life of 138.38 days, can be analyzed in isolation because of its “long” half-life. ^210^Po is built by the decay of the parent nuclides ^210^Pb and ^210^Bi. It disintegrates into the stable ^206^Pb and ends this decay chain. Polonium nuclides belong to the most powerful alpha emitters with alpha energies of 4–9 MeV. In Table 3, the naturally occurring polonium nuclides and the two artificial nuclides ^208^Po and ^209^Po, used as tracers, are listed.

The most common analytical approach for ^210^Po is its auto deposition onto a copper or silver surface in a reducing milieu. The adsorbed ^210^Po is then measured with a surface barrier detector or a gas proportional counter. ^208^Po and ^209^Po can be used as tracers. They have favorable half-lives of 2.9 years and 102 years [24,25,26,27].

### 2.1. General Methods

Case and McDowell published a new, very sensitive method for ^210^Po using an extractive scintillation cocktail. The cocktail consists of the scintillator PBBO and the extractant trioctylphosphine oxide (TOPO) [28]. These are dissolved in scintillation-grade naphthalene and toluene. This cocktail is commercially available as POLEX_a_^TM^. For the extraction, it is necessary to have set a hydrochloric acid medium. The addition of phosphoric acid suppresses the coextraction of iron and uranium. As the extraction process seems to be slow, it is important to let the extractant equilibrate for at least 20 min. In the presence of 0.1 M HCl and 7.4 M phosphoric acid, the extraction coefficients were between 10^3^ and 10^4^ [28]. The authors were successful in analyzing solid samples (ores, soils) and water samples. Monna et al. and Borysenko et al. used the same technique to analyze marine sediment samples and environmental spills at uranium mining projects, respectively [29,30]. Véronneau et al. compared the extractions with POLEX to extractions using URAEX. In the presence of sulfuric acid and sodium chloride, they obtained polonium-selective extractions with URAEX. They also observed the coextraction of plutonium, uranium and thorium. With increasing chloride concentrations, the extraction of polonium became dominant (D > 10^4^) [17]. Monna et al. reported overall recoveries for ^210^Po in sediment samples of 99.5 ± 6.5% when extracted with POLEX [29]. Among others, Landstetter et al. used POLEX for the analysis of water samples [31,32,33].

When solid samples are analyzed it is important to not heat any solution to over 200 °C. Otherwise, losses of polonium will occur. A typical dissolution process is microwave digestion in an acidic milieu together with H_2_O_2_. It is possible to use tracers, such as ^208^Po or ^209^Po. For example, Borysenko et al. used ^209^Po as a tracer for analyzing environmental spills [30].

### 2.2. Polonium Trace-Analyses

In our approach, based on the description of Case and McDowell [28], water samples were analyzed as follows: 10 mL of 25% hydrochloric acid and 100 mL of 85% phosphoric acid are carefully added to 1 L of sample in a glass bottle. Then, 5 mL of POLEX is added and the mixture is stirred vigorously with a magnetic stirrer for 5 min. The bottle is then left to stand for one hour for complete phase separation. The POLEX phase is separated from the water by means of a phase separator. The POLEX is purged with argon gas. Then, 1.5 mL is transferred to a culture tube. After setting the correct PSD (typically 2.2–2.5 ns), the pulse height spectrum is measured for 24 h (Figure 4). The recoveries in tap water were 115 ± 5% at a level of 42 mBq/L, with an overall precision of 2.5% (*n* = 7 samples).

### 2.3. Drinking Water Monitoring

During drinking water monitoring of the two Swiss states of Basel–Country and Basel–City in 2014, the local tap water from every village, in total 122 tap water samples, was analyzed. Figure 4 shows typical spectra of Po PERALS analyses. The analyses of drinking water from the village Riehen showed elevated activities of 81–200 mBq/L ^210^Po. The samples were collected at public fountains, which are supplied with local ground water. This ground water is rich in radon (40–90 Bq/L) [34]. This explains the traces of polonium found. The German reference value of 20 mBq/L ^210^Po for the preparation of infant formulas is here exceeded [2]. However, these fountains are not used for daily consumption by the public. The tap water of this village, which is supported by the City of Basel, contains 23 ± 21 mBq/L. The mean activity in the tap water of the villages of Basel–Country was 9 ± 13 mBq/L, whereas the mean of the three villages of Basel–City was slightly elevated: 16 ± 7 mBq/L ^210^Po. This is explained by the origin of the water for drinking water production. About 25% of the city’s tap water is groundwater from the village of Riehen, which contains elevated activities of radon [35].

## 3. Radium

Radium species occur in all three natural decay series. ^228^Ra (a pure beta emitter) and ^224^Ra belong to the ^232^Th decay chain. ^226^Ra is the parent of ^222^Rn and belongs to the ^238^U decay chain. ^223^Ra is a member of the ^235^U decay chain and has gained importance as a cancer agent in recent years. In drinking water, ^228^Ra and ^226^Ra are of radiological concern. In tap water and mineral waters, ^228^Ra is one of the prominent radionuclides causing relevant contributions to the indicated dose (Table 4).

### 3.1. General Methods

Case and McDowell developed an efficient extraction/scintillation system for radium. The extractant, commercially available as RADAEX, contains a neocarboxylic acid and dicyclohexano-21-crown-7 as extractants. PBBO is the scintillator [8,36].

In several publications, the conditions for the extraction of radium from water samples were investigated and optimized [30,37,38,39,40,41]. Aupiais proposed the addition of α-hydroxy-isobutyric acid to prevent salt precipitations in samples of high salinity (pH has to be set > 10 for extraction with RADAEX) [39]. Jia and Jia published a review of the analytical methods for radium in environmental samples [42]. This extraction technique is very suitable for samples wherein radium is present with elevated activity, such as soils and ores. Hashimoto et al. reported the successful determination of different radium species from ore samples using RADAEX combined with time interval analysis (TIA) and pulse shape discrimination (PSD) [43].

### 3.2. Radium Trace Analyses

The extraction of 1 L of Pedras mineral water with 5 mL of RADAEX at pH > 10, according to the ORDELA-Notes [8], resulted in a poor extraction recovery of only 4%. A 1:1 extraction with RADAEX achieved a recovery of 88% (theoretically calculated to ~92%, see Equation (2)). The distribution coefficient for radium with RADAEX is about 100 when enough nitrate is present and with a pH set > 10. A significant preconcentration of radium with RADAEX is not possible. Therefore, such extractions are not suitable for trace analyses since the achievable limit of detection (LOD) is too high.

At the state laboratory of Basel–City, radium trace analyses in water samples are performed successfully with an adsorptive alpha spectrometric method. Radium species are adsorbed onto a MnO_2_ disk following alpha spectrometry with a silicon surface barrier detector [43]. The LOD for ^224^Ra and ^226^Ra is 2 mBq/L. The resolution is sufficient to distinguish between the radium peaks and the peaks of their radon and polonium daughters [44].

### 3.3. Drinking Water Monitoring

During the tap water campaign in 2014, the state laboratory of Basel–City analyzed more than 120 tap water samples for radium. The found ^226^Ra activities were low (<2–80 mBq/L) with a mean of 6 ± 11 mBq/L compared with ^228^Ra. The latter is a dose-relevant radionuclide. The measured activities were 80 ± 101 mBq/L. ^224^Ra was found only sporadically with activities below 10 mBq/L (mean: 4 ± 3 mBq/L, *n* = 4 samples) [35] (Figure 5).

### 3.4. Radium Monitoring Campaigns

In 2014, a complete analysis of the tap water of all villages of the two states of Basel–City and Basel–Country was realized. ^226^Ra was detected in over 200 tap water samples in a range from <2 to 80 mBq/L. The mean activity in Basel–Country was 14 ± 16 mBq/L, and in Basel–City 9 ± 8 mBq/L. In the case of ^224^Ra, most samples showed activities below the detection limit of 2 mBq/L. Only in a few samples was ^224^Ra detectable. In six tap water samples of Basel–Country, ^228^Ra activity was the dominant Ra species. The mean activity was 134 ± 96 mBq/L. In conclusion, the activity of the radium species in the tap water of the two states was low and in most samples was below the corresponding DAL value (Table 1) [35].

In 2018, the state laboratory of Basel–City investigated 46 kinds of mineral water from the Swiss market. Similar to the tap water campaign, ^228^Ra was the dominant radium species. It was found in 11 mineral waters with a mean activity of 73 ± 63 mBq/L (range: <50–400 mBq/L). ^226^Ra was detectable in 23 samples, but with lower activities (44 ± 205 mBq/L, range: <2–1360 mBq/L). ^224^Ra was found in only two samples (3 and 107 mBq/L) [45] (Figure 5). The most remarkable sample was PEDRAS, a Portuguese mineral water. In this mineral water, the highest ^226^Ra activity was found, with 1360 mBq/L, as well as ^224^Ra 110 mBq/L and ^228^Ra 1600 mBq/L. Surprisingly, ^223^Ra from the ^235^U decay series was also detectable (100 ± 80 mBq/L) together with another member of this decay chain, ^227^Th (200 ± 80 mBq/L). The regular consumption of this mineral water may lead to doses of 1–10 mSv/year [46].

## 4. Radon

Each natural decay chain contains one radon nuclide. The most important is ^222^Rn of the ^238^U decay series. Its half-life of 3.8 days allows its emanation from the soil and its dispersion in the atmosphere (including the basements of buildings). Today, it is the major risk factor for lung cancer beside tobacco. Thoron, ^220^Rn, has only a short half-life of 1 min, but it can become a health risk when materials used for the construction of buildings are rich in ^232^Th/^228^Th. The permanent decay of thorium releases thoron continuously into the air. ^219^Rn is of no radiological concern, since its half-life is too short (Table 5).

### 4.1. General Methods

For radon analyses, McDowell proposes an extraction procedure for 10–15 mL water samples with water/extractant ratios of 1:1 up to 10:1. For the alpha counting, he proposed special vials with a capillary entry to prevent any losses of radon. In Application Note 7 from ORDELA Inc., 21 mL of water is extracted with 3 mL of RADONS in a 24 mL EPA vial. The achievable LOD is about 0.05 Bq/L [47].

Hamanaka et al. used 1 mL vials with a cup without any serious losses of radon. They extracted a 1 L water sample with 3 mL of a Xylene/PPO system. They reported a detector efficiency of 8.7% [48]. At the state laboratory of Basel–City, extraction experiments of ground water containing 5 Bq/L of radon were executed in accordance with the ORDELA-protocol [47]; 42 mL of ground water was extracted with 6 mL of RADONS. The extraction was repeated four times. The resulting radon activity concentrations were 2.9 ± 0.7 Bq/L with a high standard deviation of 25%, with *n* = 5. This is due to the difficult handling of the sample extraction procedure. A small air volume in the extraction vial is unavoidable. An overall recovery of 57% of the radon was achieved. This is smaller than the theoretical value of 88% (according to Equation (2) and a supposed distribution coefficient of about 50 for RADONS). The extraction of 1 L water sample with 3 mL of RADONS yielded an extraction rate of only 9%. Therefore, the used radon protocol is useful only for the extraction of small water volumes.

Many alternative methods for the determination of radon in water are well established (e.g., liquid scintillation counting (LSC)). They are easier to perform and are also more precise in repetition and more sensitive. Consequently, the following data were generated using a routine LSC method.

### 4.2. Radon Trace Analyses

At the state laboratory of Basel–City, a simple, well-established LSC method is used for radon. The water sample is collected in 20 mL scintillation vials in such a way that no air bubble remains in the vial. In the laboratory, 10 mL of water sample is mixed with 10 mL of MaxiLight cocktail in a 20 mL scintillation vial and let stand for three hours to achieve secular equilibrium. The vial is counted by means of a Hidex 300SL LSC counter for one hour. The alpha/beta discriminator is set to count only the three alpha decays of radon and its two polonium daughters. The LOD for ^222^Rn is 0.4 Bq/L.

### 4.3. Drinking Water Monitoring

During the drinking water campaign in 2014, the mean radon activity concentration of all samples was 4.9 Bq/L. The mean of the tap water in Basel–City was higher than in Basel–Country: 5.2 ± 1.4 Bq/L (3–7 Bq/L) and 4.6 ± 4.1 Bq/L (1–18 Bq/L).

This can be explained by the drinking water production situation in Basel–City. Radon-rich groundwater, which is enriched by infiltration of river Rhine water, is the source of the drinking water. Elevated radon activities were found in four fountains in the village Riehen. These fountains are recharged by radon-rich groundwater [35] (Figure 6).

## 5. Strontium

The pure beta-emitters ^89^Sr and ^90^Sr are very important fallout nuclides. In particular, the long-lived ^90^Sr and its daughter ^90^Y are of major concern (Table 6). Global bomb fallout has caused widespread contamination of the northern hemisphere. In 1963, a maximum of 97 PBq (2.62 MCi) was deposited in the northern and 12 PBq (0.31 MCi) in the southern hemisphere, respectively. In 1976, Feely estimated the cumulated radiostrontium deposition to be 315 PBq (8.5 MCi) in the north and 104 PBq (2.8 MCi) in the south [49].

### 5.1. General Methods

This widespread contamination with this dangerous radionuclide necessitates fast and effective analytical techniques. One technique includes the isolation of radiostrontium by means of ion exchange resins and the detection of ^89^Sr and ^90^Sr with liquid scintillation. Other techniques use the more sensitive approach of detecting the daughter ^90^Y. This can be isolated from ^90^Sr by precipitation or via chromatographic separation (ion chromatography) and detected with liquid scintillation detectors (even Cherenkov counting is possible). Today, many analytical techniques are based on the work of Dietz and Horwitz [50]. All these techniques, though, even when titled “fast methods”, are laborious and time-consuming.

### 5.2. Strontium Trace Analyses

In 1995, McDowell proposed an approach using selective liquid extraction of radiostrontium by means of a crown ether. After investigating several crown ethers, dibenzo-18-crown-6 (DC18C6) proved to be the most appropriate one. Combined with the scintillator didodecyl naphthalene sulfonic acid (HDDNS) solved in toluene, this extractant showed a very favorable partition coefficient of about D = 6000 for radiostrontium in water [51,52].

On the basis of the analytical scheme of McDowell [53], the state laboratory of Basel–City developed a rapid method for the analyses of water samples suitable for tap water, mineral water, river water and even sea water [20]. A 1 L sample was filtered into a clean glass bottle to remove particles. The pH was adjusted to 10–10.5 with 1 M NaOH solution. Then, 8 mL of preconditioned STRONEX extractant was added and stirred vigorously for 5 min. When adjusting the pH, hydroxide precipitations of calcium and magnesium may occur (the magnesium concentration in sea water is high). This leads to substantial losses of strontium. Because this precipitation process is slow, one possibility to overcome the problem of losses is to adjust the pH and immediately add the Stronex for the extraction. Another possibility is the dilution of samples (e.g., sea water had to be diluted 1:10 to prevent the precipitation of Mg(OH)_2_). With this fast method, it is possible to analyze at least 12 samples per working day by means of one LSC counter (Figure 7) [20]. The phases were allowed to separate for one hour. The upper scintillator phase was removed using a micro separator system. Sometimes, emulsions occurred without any clear phase separation. Water droplets were then removed by centrifugation (5 min, 2000 rpm). The STRONEX was transferred into a scintillation vial and counted with an α/β liquid scintillator counter for one hour. STRONEX contains a lipophilic acid and is purchased in the stable acidic form. Before the extraction, this acid has to be converted into the ionic form by mixing STRONEX with a mixture of NaNO_3_/NaOH 1:1 (*v*/*v*) (0.3 M and 0.2 M, respectively).

The method shows a good linearity within the working range of 0.1–1000 Bq/L (Figure 7). The recoveries for tap water and river water were 80–100%. River water with a higher hardness showed lower recoveries of 45–80%. In sea-water, the samples had to be diluted 1:10 with distilled water to avoid the precipitation of magnesium hydroxide. Recoveries of 92 ± 5% with a detection limit of 1 Bq/L can be achieved.

A synthetic water sample from an interlaboratory test of the Bundesamt für Strahlenschutz, Berlin, in 2013 gave good recoveries of 18.2 ± 4.0 Bq/L (the reference activity was 19.2 Bq/L). The sample also contained other radionuclides, such as ^60^Co, ^137^Cs, ^139^Ce and ^152^Eu.

In conclusion, the presented procedure is a suitable alternative to the more common methods. To avoid high costs (STRONEX is an expensive chemical), a simple method for the cleaning of STRONEX for reuse was developed [20].

## 6. Thorium

The three natural decay series contain six thorium species, but only three species are important for alpha analyses. ^232^Th and ^228^Th belong to the ^232^Th decay series and are long-lived nuclides. The half-life of ^228^Th is only 1.92 years and it is supported by the decay of its parent, ^232^Th. ^230^Th belongs to the ^238^U decay chain and is supported by the decay of its long-lived parent ^234^U (Table 7).

### 6.1. General Methods

McDowell developed an extractive scintillation cocktail for thorium analyses in phosphate fertilizers on the basis of trioctylphosphine oxide (TOPO) and 1-nonyldecylamine sulphate as extractants and PBBO as scintillator [8]. Dacheux and Aupiais reported the quantitative analyses of French mineral waters Badoit and Volvic for actinides. They achieved recoveries of 99 ± 4% for thorium using THOREX in 2M sulfuric acid system with a 10:1 (*v*/*v*) ratio water/THOREX [54]. Wallner analyzed thorium in urine with THOREX [55].

Ayranow et al. [56] and Füeg et al. [57] reported sediment and soil analyses for thorium and other actinides. They achieved good separations using extraction chromatography (U/TEVA) for the separation of U and Th. Fourest et al. performed a comparison for the analyses of Th with ICP-MS, PIXE and Perals. All three types of analyses were in good agreement when used for the determination of the solubility of thorium phosphate/diphosphate [58].

### 6.2. Thorium Trace Analyses

At the state laboratory of Basel–City, an analytical method for the determination of thorium traces in water samples was developed on the basis of McDowell’s work.

A 1 L water sample was transferred into a cleaned glass bottle and 100 mL of 96% sulfuric acid was carefully added. After the cooling of the mixture, 5 mL of THOREX was added and stirred vigorously for 5 min. The mixture was rest for at least 1 h to achieve a complete phase separation. The THOREX phase was removed from the water phase by means of a phase separator. The THOREX was purged for 1 min with argon gas, then 1.5 mL was transferred into a PERALS vial and counted on a PERALS counter for 24 h. Even after a rigorous clean of the glassware with 2M hydrochloric or nitric acid, a small degree of background activity remains. For ^228^Th and ^230Th^, it was calculated as 1 mBq/L, and was 2 mBq/L for ^232^Th, when extracting a 1 L sample and counting 1.5 mL of a 5 mL THOREX extract for 24 h. The recoveries were quite satisfactory: 96 ± 24% was achieved for ^228^Th in tap water and 104 ± 10% in mineral water, respectively. For ^232^Th, recoveries of 105 ± 16% in tap water and 96 ± 8% in mineral water, respectively, were obtained (Figure 8).

### 6.3. Thorium in Mineral Waters

Only traces of thorium were found in 46 mineral waters from the Swiss market (mean sum: 8 mBq/L). Thorium nuclides are particle-bound and therefore removed by the production of the mineral waters [45]. The Pedras mineral water contained 200 ± 80 mBq/L of ^227^Th, which was analyzed with gamma spectrometry. Even when thorium nuclides are particle-bound and removed mostly by the drinking water production, the analysis for thorium completes the survey of natural radionuclides.

## 7. Uranium

Three uranium species appear in the two decay chains of ^235^U and ^238^U. All nuclides are very long-lived and are alpha emitters. Special nuclides, such as ^233^U or ^236^U, are of less importance in water analysis. The artificial nuclide ^232^U is often used as a tracer (Table 8).

### 7.1. General Methods

Leyba et al. described a rapid method for the determination of uranium in water samples. They extracted a 100 mL water sample with URAEX and sulfuric acid at a pH of 2 in a separatory funnel. The counting times were a maximum of 120 min. The pH should not be lower than 2 [59]. Dacheux and Aupiais used an analytical scheme for the determination of Th, U, Pu, Am and Cm in water samples. Th and U were extracted with ALPHAEX. Uranium was then separated from Th via extraction with URAEX/H_2_SO4. The method was successful for mineral waters (Badoit) and urine [54]. Duffey et al. developed a new standard method for U in drinking water. They extracted the water sample with DTPA and extracted U with ALPHAEX. They achieved considerable decontamination factors of U from Am, Po, Pu, Ra and Th, and proposed the method as a new ASTM standard method for U in drinking water [60]. Aupiais eliminated actinides via complexation with DTPA and extracted U with ALPHAEX. He successfully analyzed reference water samples from IAEA [61]. The published ASTM standard method is D6239, describing excellent recoveries in tap water (95–105%) and an LOD of 37 mBq/L [62]. Dacheux et al. compared extractions of ALPHAEX/PERALS with Ultima Gold LLT/TRI-CARB extraction/detection systems for the determination of actinides. The PERALS showed the best resolution of the uranium nuclides ^232^U, ^234^U and ^238^U [63]. Hinton described an analytical method for the determination of uranium in urine. After a clean-up of the urine extract over a strong anion exchange column (AG1-X8), the extracts were either dried on a planchet and analyses by means of a gas flow proportional counter or ALPHAEX extracts were analyzed with PERALS [64,65]. Dinse et al. used a combined analytical scheme for the analysis of Pu, Am and U in urine. Am was analyzed with PBBO as a scintillator using the PERALS system [66].

Several authors described the analysis of uranium in solid samples. Ayranov analyzed marine sediments with an extracting schema using URAEX and THOREX for the analysis of U and Th after microwave digestion of the sample with nitric acid and hydrogen peroxide [56]. Füeg et al. analyzed soils after a separation of U and Th with a U/TEVA resin. They used URAEX and THOREX. Radium was also analyzed using RADEX [57]. Borysenko analyzed environmental spills at a mine with URAEX for uranium [30].

### 7.2. Uranium Trace Analyses

On the basis of McDowell’s propositions for the extraction of uranium with URAEX [8], a sensitive method for the analysis of uranium in water was developed at the state laboratory of Basel–City: 20 mL of 96% sulfuric acid was carefully added to a one-liter water sample in a pre-cleaned glass bottle. Water containing carbon dioxide has to be stirred for some time to outgas any CO_2_, otherwise losses during the extraction process are unavoidable. Then, ^232^U as tracer and 5 mL of URAEX were added. The mixture is adjusted with deionized water to about 1 L and stirred vigorously for five minutes with a magnetic bar. A good separation of the URAEX layer on top of the water is achieved after one hour. The URAEX layer is separated by means of a phase separator. The URAEX is purged with argon for 30 s, and 1.5 mL of URAEX is pipetted into a PERALS cuvette and counted for 24 h after setting the PSD (Figure 9).

Even after a rigorous cleaning of the glassware with acid, a minimal background level of 1–3 mBq/L of ^234^U and ^238^U remains. This has to be considered when calculating the activities. The recoveries for mineral waters were between 80 and 122% (101 ± 30%, at a level of 50 mBq/L) after correction with the recovery of the tracer ^232^U. Similar recoveries were achieved in Swiss tap waters: 65–112% (74 ± 24%, level: 50 mBq/L). The detection limit was 4 mBq/L for the 500 mL sample with a counting time of 24 h.

### 7.3. Uranium Monitoring Campaigns

During the last 20 years, the state laboratory Basel–City has undertaken many investigations into the uranium in drinking water, mineral waters and river water. The most important factor was the evidence that only a thorough pre-cleaning of the glassware allows one to avoid cross-contaminations (see Chapter 1). In 2014, the state laboratory analyzed the tap water of all water production sites of the cantons of Basel–Country and Basel–City. The level of Uranium found was well below the limit of 30 µg/L rsp. 2.8 and 3.0 Bq/L for ^234^U rsp. ^238^U. The mean values of the two states were comparable: 21 ± 28 mBq/L in Basel–Country and 26 ± 4 mBq/L in Basel–City. The data from the city showed less variation than in the country, because the city has only one drinking water production plant, which uses groundwater from two production sites. The variation in the uranium activity in the tap waters in Basel–Country ranges from 2 to 120 mBq/L, and refers to the tap water of 89 villages [35] (see Figure 9).

In 2018, 46 kinds of mineral water, which are available on the Swiss market, were investigated. In 45 samples Uranium was detectable. The mean was 1.9 ± 2.0 µg/L, with a maximum of 11 µg/L in an Italian mineral water. All in all, we could not detect any severe contamination of mineral waters with Uranium [45] (see Figure 9).

## 8. Further Actinides

Dacheux and Aupiais proposed a sequential extraction scheme for the analyses of actinides in mineral waters. Uranium and thorium are separated with ALPHAEX/H_2_SO_4_ extraction. Pu, Am and Cm are extracted with ALPHAEX/HNO_3_. U and Th are separated by sequential extraction with URAEX and THOREX, respectively. With ALPHAEX/HNO_3_ (pH = 1), plutonium can be extracted from the remaining water sample. Americium and curium can then be extracted separately after a complicated oxidation process followed by HDEHP–Teflon chromatography [54]. In the paper of Dacheux and Aupiais (1998), the separation of americium from curium is explained in detail. As the process is complicated, losses are unavoidable. The use of tracers, such as ^246^Cm or ^248^Cm for ^244^Cm and ^148^Gd for ^241^Am and ^244^Cm, is proposed [67].

Xia et al. extracted plutonium from compacted bentonite with hydrochloric acid. The filtered acid was mixed with the scintillator Ultima Gold AB and measured on a PERALS [68]. Zapta-Garcia et al. analyzed the uranium and plutonium in contaminated concrete from Sellafield. They used a combination of the ion exchange columns AG1-8X and TEVA/UTEVA and extractants URAEX/ALPHAEXD for this purpose [69].

Neptunium is also of interest. Aupiais et al. optimized an extraction/detection procedure for neptunium in biological samples (liver and femur of rats), as well as other actinides. Extraction was performed with the TOPO-containing POLEX/HNO_3_ after changing all neptunium to the +6 state [70]. Baglan et al. proposed three different approaches for the sequential extraction/determination of actinides. They simplified the procedure after avoiding the total separation of each actinide. One extract contained uranium and thorium, the other extract neptunium and plutonium. The measurement of these extracts was possible with ICP/MS [71].

## 9. Conclusions

PERALS alpha spectrometry is a useful technique in the analyses of alpha nuclides in water samples. Over the last 20 years, a lot of experience has been gained in optimizing this technique and applying it in the routine monitoring programs of the state laboratory of Basel–City, especially in polonium, thorium and uranium analyses. In the opinion of the authors, the pivotal point is the avoidance of contamination in sample preparation. This implies the use of chemicals with the highest purification grade and a thorough cleaning of the used glassware. The commercially available cocktails from ORDELA Inc. are of the purest quality (scintillation grade), which explains their price. The consumption of cocktails is limited. The analyzed cocktail solutions may be cleaned and used for extraction again.

The analysis of solid samples, e.g., vegetables, fruit, fish or honey, is also possible. First, samples have to be ashed (calcination) or mineralized (e.g., using a microwave digestion). The ashes are dissolved in concentrated acid, diluted with distilled water and extracted analogously to a water sample. Because the amount of the ashed material is limited, the achievable LODS are higher. Trace analysis is not possible when analyzing such samples with PERALS.

## Figures and Tables

**Figure 1 materials-14-03787-f001:**
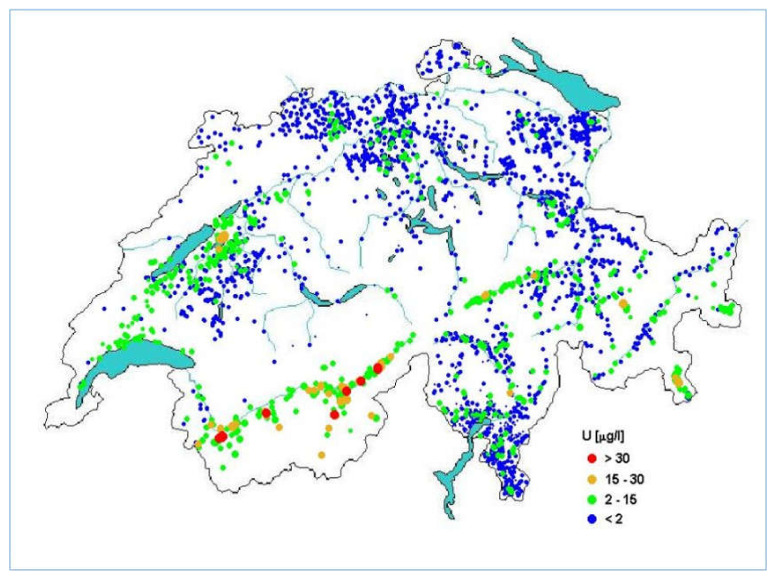
Concentrations of Uranium in Swiss tap water (µg/L). Taken from *Natürliche Radioaktivität im Schweizer Trinkwasser* (published in [7] with kind permission of Heinz Surbeck).

**Figure 2 materials-14-03787-f002:**
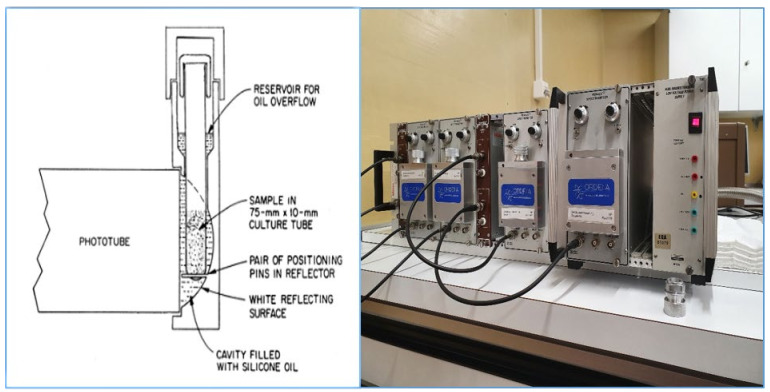
(**Left**): Scheme of the PERALS cell with the photo tube, ORDELA model 8100AB-HV (taken from [8] with kind permission of ORDELA Inc.). (**Right**): A series of four PERALS counters (photo by M. Zehringer).

**Figure 3 materials-14-03787-f003:**
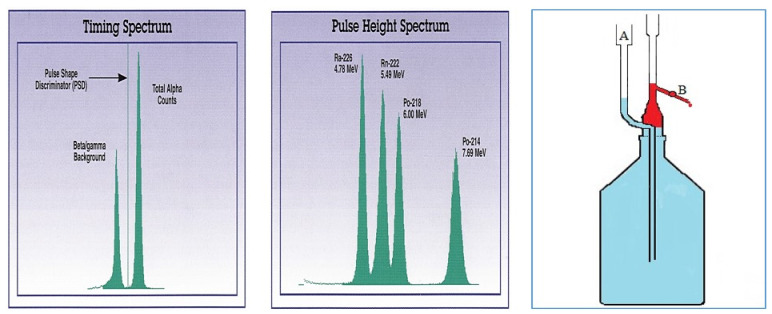
(**Left**): Pulse shape spectrum of a ^226^Ra standard (78 Bq). The PSD is set left from the alpha peak. Middle: Pulse height spectrum of the same ^226^Ra standard with daughter nuclides in equilibrium (taken from [19] with kind permission of ORDELA Inc). (**Right**): Extraction bottle with phase separator (red: separated cocktail phase) [20].

**Figure 4 materials-14-03787-f004:**
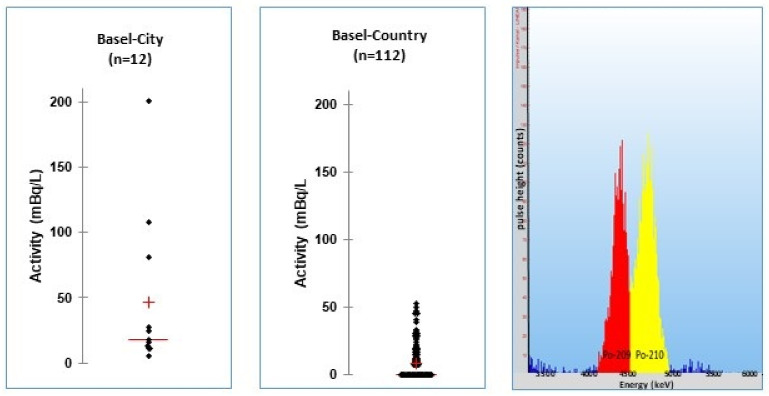
(**Left**,**middle**): Distribution of the ^210^Po activities in the tap water of the villages of the two states Basel–City and Basel–Country. (**Right**): PERALS spectrum of a ^210^Po standard with its tracer ^209^Po (each about 100 mBq/L).

**Figure 5 materials-14-03787-f005:**
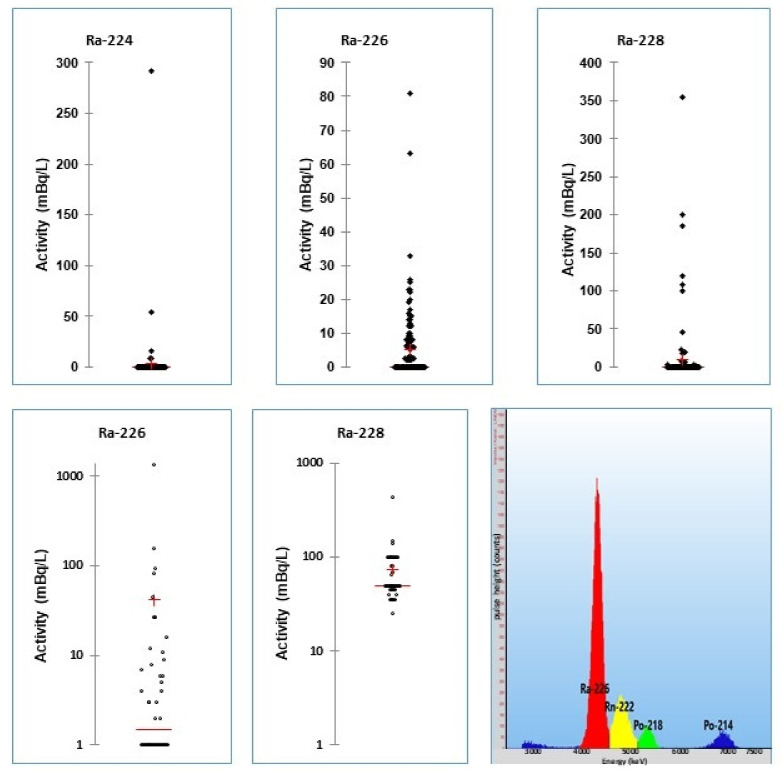
Upper diagrams: scatter diagrams of radium nuclides in tap water. Lower diagrams left and middle: scatter diagrams of radium activities in 46 mineral waters from the Swiss market (from [35]). Right: PERALS spectrum of a RADAEX extract of the mineral water PEDRAS, showing ^224^Ra, ^226^Ra and daughters. The PEDRAS mineral water contains about 1.4 Bq/L ^226^Ra and 110 mBq of ^224^Ra [46].

**Figure 6 materials-14-03787-f006:**
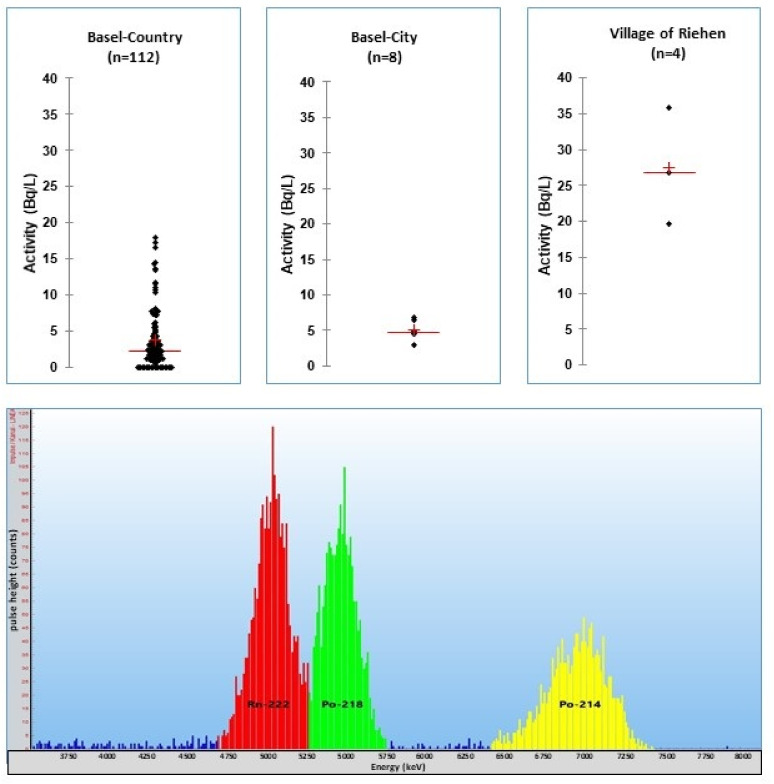
Scatter diagrams of radon activity concentrations in 124 tap water samples of all villages in Basel–Country, Basel–City and the village of Riehen [45]. Spectrum below: RADONS extract of a ground water sample (village Riehen, 5 Bq/L ^222^Rn).

**Figure 7 materials-14-03787-f007:**
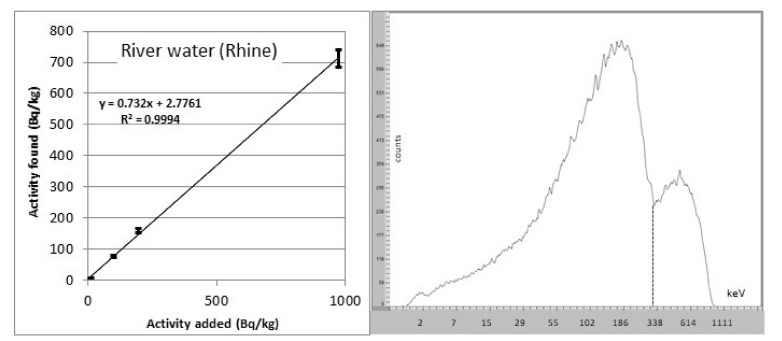
Left: linearity function of ^90^Sr LSC measurements in river water (source: [20]). Right: ^90^Sr spectrum counted on a Hidex LSC counter. The spectrum is strongly quenched.

**Figure 8 materials-14-03787-f008:**
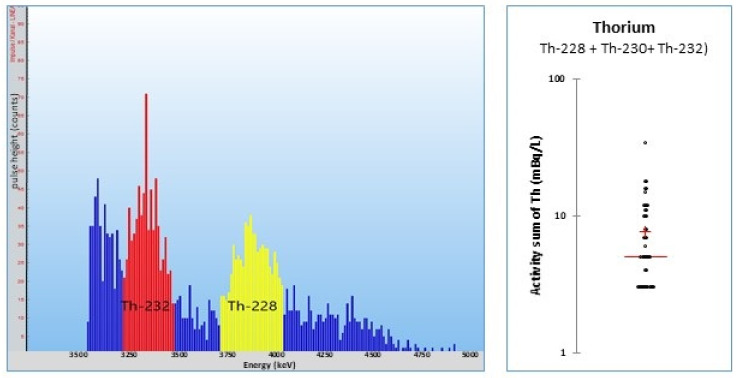
Left: THOREX extract of a ground water sample. Right: scatter diagram of the activity sum of the three thorium nuclides in mineral water samples [45].

**Figure 9 materials-14-03787-f009:**
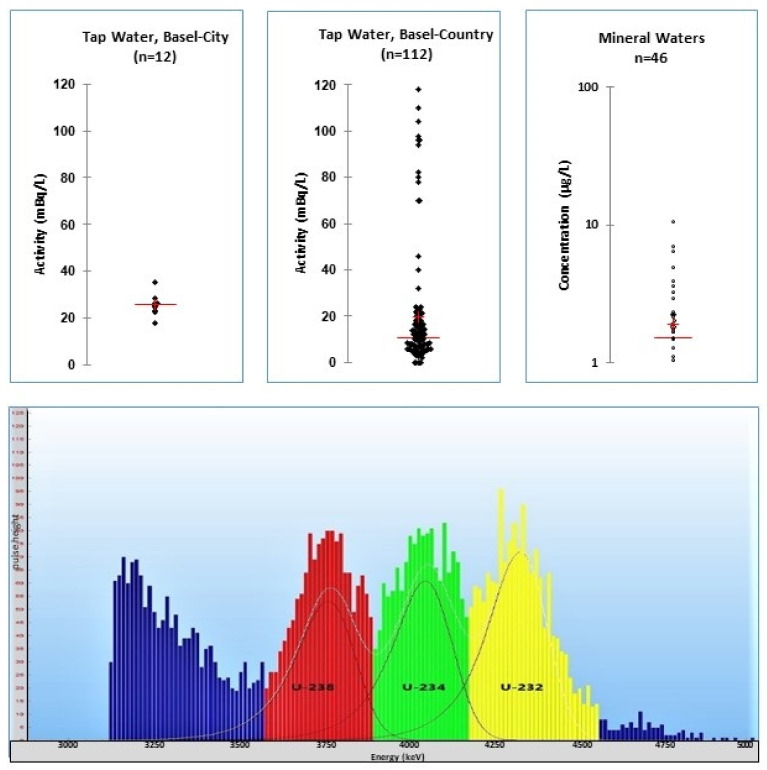
Upper scatter diagrams: tap waters of Basel–City and Basel–Country [35] and mineral waters from the Swiss market [45]. PERALS spectrum of the mineral water *San Pellegrino*. Note the quenching of more than 500 keV. The found activities were 82 mBq/L ^234^U and 76 mBq/L ^238^U. Added tracer ^232^U: 100 mBq.

**Table 1 materials-14-03787-t001:** Activity levels for drinking water in Switzerland and corresponding ingestion factors. DAL: derived activity limits calculated with Equation (1). * Dose coefficients for ingestion listed in the Swiss Radiological Protection Ordinance [5] based on ICRP-Publication 119 [1]. EU: Council Directive 2013/15/EURATOM [4].

Drinking Water				Dose Coefficients for Different Age Groups *		
Radionuclide	TBDV [3]	DAL Infant	DAL Adult	Infant (1–2 Years)	Children >10 Years	Adult Person
	Bq/L	Bq/L	Bq/L	µSv/Bq	µSv/Bq	µSv/Bq
Americium-241 (^241^Am)	0.1	0.4	0.7	0.37	0.22	0.20
Lead-210 (^210^Pb)		0.04	0.2	3.6	1.9	0.69
Polonium-210 (^210^Po)		0.02	0.1	8.8	2.6	1.2
Plutonium ^239 + 240^Pu		0.3	0.6	0.42	0.27	0.25
Radium-224 (^224^Ra)		0.2	2	0.66	0.26	0.07
Radium-226 (^226^Ra)		0.1	0.5	0.96	0.8	0.28
Radium-228 (^228^Ra)		0.1	0.2	30	3.9	0.69
Radon (^222^Rn)	100	2	5	0.073	0.06	0.028
Strontium-90 (^90^Sr)	4.9	0.4	2	0.37	0.14	0.07
Thorium-228 (^228^Th)		0.4	2	0.37	0.14	0.07
Thorium-230 (^230^Th)		0.3	0.6	0.41	0.29	0.23
Thorium-232 (^232^Th)		0.3	5	0.45	0.06	0.028
Uranium-234 (^234^U)		1	2.8	0.13	0.074	0.049
Uranium-238 (^238^U)		1	3	0.12	0.068	0.045

**Table 2 materials-14-03787-t002:** Extractive cocktails for the PERALS methods (taken from [8]). PBBO: 2-(4′-biphenylyl)-6-phenylbenzoxazole, PPO: 2,5-diphenyloxazole, NPO: 2-(1-Naphthyl)-5-phenyloxazole, HDEHP: bis-(2-ethylhexyl)phosphoric acid, TOPO: Trioctylphosphine oxide.

Cocktail	Scintillator	Extractant	Energy Transfer Agent	Solvent
ALPHAEX	PBBO, PPO, NPO	HDEHP	Naphthalene	Toluene
POLEX	PBBO	TOPO	Naphthalene	Toluene
RADAEX	PBBO	2-methyl-2-heptyl-nonanoic acid, Dicyclohexano-21-crown-7	Naphthalene	Toluene
RADONS	PBBO	No extractant	Naphthalene	Xylene
STRONEX	PBBO	C6-C8-Neocarboxilic acids, Dicyclohexano-18-crown-6	Naphthalene	Toluene
URAEX	PBBO	Tert. Amine (MG > 300)	Naphthalene	Toluene
THOREX	PBBO	Primary Amine (MG > 300)	Naphthalene	Toluene

**Table 3 materials-14-03787-t003:** Properties of polonium nuclides of the natural decay series and important tracers. Alpha energy lines in keV with emission probability in %. All data from [22,23].

Polonium Nuclide	Decay Chain	Decay	Half-Life	Main Alpha Energy Lines
^218^Po	^238^U-series	α	3.1 m	6002 (100)
^214^Po		α	162 µs	7687 (100)
^210^Po		α	138.4 d	5304 (100)
^216^Po	^232^Th-series	α	0.15 s	6778 (100)
^212^Po		α	0.3 µs	8785 (100)
^215^Po	^235^U-series	α	1.8 µs	7386 (100)
^211^Po		α	0.52 s	7450 (99)
^208^Po	tracer	α	2.99 y	5216 (100)
^209^Po	tracer	α	102 y	4977 (79) 4979 (20)

**Table 4 materials-14-03787-t004:** Properties of naturally occurring radium nuclides. Alpha energy lines in keV with emission probability in %. All data from [22].

Radium Nuclide	Decay	Half-Life	Main Alpha Energy Lines
^228^Ra	β	5.75 y	
^226^Ra	α	1600 y	4601 (6), 4784 (94)
^224^Ra	α	3.63 d	5448 (5), 5685 (94)
^223^Ra	α	11.43 d	5539 (11), 5607 (26) 5713 (50), 5757 (10)

**Table 5 materials-14-03787-t005:** Properties of radon nuclides. Alpha energy lines in keV with emission probability in %. All data from [22].

Radon Nuclide	Decay Chain	Decay	Half-Life	Main Alpha Energy Lines
^222^Rn (“radon“)	^238^U-series	α	3.82 d	5489 (100)
^220^Rn (“thoron”)	^232^Th-series	α	55.8 s	5449 (5) 6288 (95)
^219^Rn (“actinon”)	^235^U-series	α	3.98 s	5425 (8), 6553 (13) 6819 (79)

**Table 6 materials-14-03787-t006:** Properties of radiostrontium and daughter nuclides. Beta energies in keV. All data from [22].

Radio Nuclide	Decay	Half-life	Energy Beta Max
**^89^Sr**	β,γ	50.6 d	1495
**^90^Sr**	β	28.8 y	545.9
**^90^Y**	β	64.1 h	2279

**Table 7 materials-14-03787-t007:** Properties of thorium nuclides of the natural decay series and the tracer ^229^Th. Alpha energy lines in keV with emission probability in %. All data from [22].

Thorium Nuclide	Decay Chain	Decay	Half-Life	Main Alpha Energy Lines
^234^Th	^238^U-series	β	4.5 10^9^ y	–
^230^Th		α	7.5 10^4^ y	4621 (23.4) 4787 (76.3)
^232^Th	^232^Th-series	α	1.4 10^10^ y	3949 (21.0) 4011 (78.9)
^228^Th		α	1.92 y	5340 (26.0) 5423 (73.4)
^231^Th	^235^U-series	β	25.52 h	–
^227^Th		α	18.72 d	5757 (20.4) 5978 (23.5) 6038 (24.2)
^229^Th	tracer	α	7.9 10^3^ y	4837 (58.2) 4894 (10.7) 4806 (11.4)

**Table 8 materials-14-03787-t008:** Properties of uranium nuclides of the natural decay series and the tracer ^232^U. Alpha energy lines in keV with emission probability in %. All data from [22].

Uranium Nuclide	Decay Chain	Decay	Half-Life	Main Alpha Energy Lines
^238^U	^238^U-series	α	4.5 10^9^ y	4151 (22.3) 4198 (77.4)
^234^U		α	2.5 10^5^ y	4722 (28.4) 4775 (71.4)
^235^U	^235^U-series	α	7.0 10^8^ y	4366 (18.8) 4398 (57.2)
^232^U	Tracer	α	72 y	5260 (32) 5320 (68)

## Data Availability

The data presented in this stury are available on request from the corresponding author.

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
