# Peer review of "Applications of PERALS-Alpha Spectrometry for the Investigation of Radionuclides in Water Samples"

_materials, 2021, doi:10.3390/ma14143787_

Round 1

Reviewer 1 Report

See attached pdf.

Author Response

Figure 7: we set a energy-scale in keV. The spectrum has a quench of about 60%, shifting peaks to lower energies. The spectrum shows Sr-90 and a small ingrowth of Y-90. Hidex corp. gave us the formula to calculete from channels to energy. According to Hidex the spectrum is typical for strontium-90 with ingrowth of its daughter Ytrium-90.

We changed "mother Isotope or nuclide" with "parent"

Fig. 4,5, 6,8 and 9: we overlayed the x-axis with keV-scale and the y-axis with pulse height (counts) to make fig. more readable. Unfortunately the spectrometry-software has poor grafical possibilities.

We considered all minor comments (many thanks to the reviewer!)

Reviewer 2 Report

The manuscript describes several protocols to determine the presence of several radionuclides in water samples. Special focus is placed to the PERALS Alpha spectrometry detection technique, applied to the determination of several alpha emitting radionuclides in water samples in Switzerland.

It is an interesting and clearly written paper, providing a useful explanation on the working details of PERALS, the experimental protocols to extract the radionuclides of interest from the samples, and the global picture of the presence of alpha emitting radionuclides in water samples in Switzerland. From my point of view this work perfectly fits the scope of the special issue "Detectors for Assessment of Natural Radioactivity in Drinking Water: Materials and Method" in the Materials Journal, although there are several points that I think the authors should address before this manuscript can be published:

1. The content of the manuscript, as interesting as it is, does not fit with one would expect from its title and abstract. Although the title and abstract talk only about PERALS as the detecting technique, most of the results presented use also other detection techniques (regular LSC, alpha spectrometry). Also, most of the paper is devoted to providing results about determination of the presence of natural radionuclides in water samples in Switzerland, but there no reference at all about these results neither in the title nor in the abstract. While I think that both the contents of this work and the way it is written is adequate, I would suggest the authors to rewrite the abstract and probably also the title to provide a more realistic description on the contents of this work.

2. It seems a little odd to start the introduction  with a review of the legal bases in Switzerland, specially when nothing has been said before about that country and in the contents there are also references to EURATOM and German regulations.

3. The title, abstract and introduction seem to point out that the focus of the paper is on the determination of alpha emitting radionuclides, but a section is devoted to Strontium. Although I think that this is an interesting section, which should be kept in the manuscript, I would suggest to place it at the end of the paper, instead as just in the middle of other sections devoted to mainly beta emitting radionuclides.

I do also have some additional minor comments:

4.  Line 52: "Figure 1 shows uranium in drinking water..." should be "Figure 1 shows concentrations of uranium in drinking water...".

5. Line 143: "(Figure 3, left)", I think should be "(Figure 3, right)".

6. Lines 229-30: "After setting the correct PSD (typically 2.2 - 2.5),...", in which units?

7. In several points of this work, recoveries of extraction for several radionuclides are reported providing reference to equation (1), instead of equation (2), which is the actual equation providing such values.

8. Line 335: "Hamanada et al. used one ml vials with cup..." should be "Hamanada et al. used 1 mL vials with cup...".

9. Line 350: "Consequently, the following date were generated using a routine LSC-method" should be "Consequently, the following data were generated using a routine LSC-method".

In summary, I think that this is an interesting paper which perfectly fits the scope of this special issue, and I do recommend its publication in the Materials journal, provided that the authors consider my previous comments, specially the rewriting of the abstract to provide a more accurate description of the actual contents of this work.

Author Response

  1. We have rewritten the abstract to get a better summary of the paper.
  2. The Swiss legislation is always harmonized to the EU-legislation. We wanted to start with the legislation as a basis for our work and the reason why we analyse drinking water.
  3. We put the radionuclides in alphabetic order. We think this gives no confusion to the reader.
  4. -9. we all considered the remarks and changed the text.